The role of macrobiota in structuring microbial communities along rocky shores

Pfister Catherine A. 1 cpfister@uchicago.edu
Gilbert Jack A. 1 2
Gibbons Sean M. 2 3
1 Department of Ecology and Evolution, University of Chicago , Chicago, IL , USA
2 Institute of Genomic and Systems Biology, Argonne National Laboratory , Lemont, IL , USA
3 Biophysical Sciences Graduate Program, University of Chicago , Chicago, IL , USA
Toonen Robert
Electronic publication date: 2014 Oct 16
Publication date: 2014
Volume: 2
Electronic Location ID: e631
Received 2014 Aug 27; Accepted 2014 Sep 30
Copyright: © 2014 Pfister et al.
Copyright year: 2014
Copyright holder: Pfister et al.
License: This is an open access article distributed under the terms of the Creative Commons Attribution License, which permits unrestricted use, distribution, reproduction and adaptation in any medium and for any purpose provided that it is properly attributed. For attribution, the original author(s), title, publication source (PeerJ) and either DOI or URL of the article must be cited.
License URL: https://creativecommons.org/licenses/by/4.0/

Keywords: 16S, Rocky intertidal, Mytilus californianus, Nitrogen cycling, Tatoosh Island, Nitrification, Animal excretion, Tidepool, Ammonium, Host-microbe

Funding: NSF OCE-0928232 Earth Microbiome Project EPA STAR Fellowship National Institutes of Health Training 5T-32EB-009412 U.S. Department of Energy DE-AC02-06CH11357 Funding by NSF OCE-0928232 (CAP), the Earth Microbiome Project (JG), an EPA STAR Fellowship (SG), and a National Institutes of Health Training Grant (to SG; 5T-32EB-009412) made this research possible. This work was supported in part by the U.S. Department of Energy under Contract DE-AC02-06CH11357. The funders had no role in study design, data collection and analysis, decision to publish, or preparation of the manuscript.

==============================
Rocky shore microbial diversity presents an excellent system to test for microbial habitat specificity or generality, enabling us to decipher how common macrobiota shape microbial community structure. At two coastal locations in the northeast Pacific Ocean, we show that microbial composition was significantly different between inert surfaces, the biogenic surfaces that included rocky shore animals and an alga, and the water column plankton. While all sampled entities had a core of common OTUs, rare OTUs drove differences among biotic and abiotic substrates. For the mussel Mytilus californianus, the shell surface harbored greater alpha diversity compared to internal tissues of the gill and siphon. Strikingly, a 7-year experimental removal of this mussel from tidepools did not significantly alter the microbial community structure of microbes associated with inert surfaces when compared with unmanipulated tidepools. However, bacterial taxa associated with nitrate reduction had greater relative abundance with mussels present, suggesting an impact of increased animal-derived nitrogen on a subset of microbial metabolism. Because the presence of mussels did not affect the structure and diversity of the microbial community on adjacent inert substrates, microbes in this rocky shore environment may be predominantly affected through direct physical association with macrobiota.

Introduction

The dynamics and interactions of the macroscopic species on rocky shores of the northeast Pacific Ocean have been well-characterized and thus have contributed significantly to our understanding of coastal ecological processes (e.g., Paine, 1966; Wootton, 1994; Estes & Duggins, 1995). Although some specialized symbiotic associations have been described in rocky shore species (Secord & Augustine, 2000; Bergschneider & Muller-Parker, 2008), we know little about multi-taxa microbial associations. There is increasing evidence that many marine macrobiota have surface biofilms (Grossart et al., 2005; Kvennefors et al., 2012) or endosymbionts (Zurel et al., 2011; Wegner et al., 2013) or both (e.g., Qian et al., 2006; Taylor et al., 2007). However, our understanding of the specificity of these associations and their functional significance remain nascent with some notable exceptions (Webster & Taylor, 2012; Fan et al., 2013; Heisterkamp et al., 2013). The shelf waters of the California Current Large Marine Ecosystem (CCLME) maintain diverse and unique microbial communities across upwelling areas (Bertagnolli et al., 2011). The relatively high productivity of this system has been attributed to the seasonal upwelling of nitrate (Barber & Smith, 1981), which can lead to significant levels of carbon fixation. The diversity of potential plant and animal ‘host’ species in the CCLME, their relatively large geographic range, their longevity and the provision of key resources suggest they provide unique microniches capable of increasing the diversity and function of microbial communities.

If macrobiota directly provide habitat for intertidal microbes (e.g., mussels Pfister, Meyer & Antonopoulos, 2010; algae, Miranda et al., 2013) or indirectly provide resources such as nitrogen, in the form of animal-regenerated ammonium, then our understanding of coastal biogeochemistry is incomplete without considering the contribution of macrobiota. In the northeast Pacific, Tatoosh Island, WA shows persistent shoreward peaks of ammonium (Pfister, Wootton & Neufeld, 2007), while the tidepools at Second Beach, WA have animal-regenerated ammonium from the California mussel with locally enhanced algal productivity, and microbial nitrification (Pfister, 2007). We employed 16S rRNA V4 amplicon sequencing to test whether microbes associated with macrobiota differ from those in the water column and on inert surfaces. We further compared the identified microbial taxa with those found when using shotgun metagenomics in a subset of samples for mussel shell biofilms. Finally, we used a manipulative field experiment of mussel presence or absence in tidepools to ask whether the increased rates of nitrogen remineralization and uptake demonstrated with mussels by tracer ammonium addition (Pather et al., 2014) resulted in changed bacterial communities on inert surfaces.

Materials and Methods

We asked whether microbial diversity and abundance were distributed differentially among intertidal microhabitats by (1) sampling distinct microhabitat types including artificial and natural surfaces at Tatoosh Island (48.32°N, 124.74°W), and (2) sampling artificial and natural substrates in the context of an animal removal experiment at nearby Second Beach (48°, 23′N, 124, 40′W). Tatoosh Island is 0.7 km off the northwestern tip of Washington State, USA and has been well studied ecologically. Previous metagenomic analysis of biofilms associated with the shells of mussels at Tatoosh Island demonstrated a rich microbial assemblage, with the genetic capacity for nitrogen cycling (Pfister, Meyer & Antonopoulos, 2010). On 6 Aug 2009, we sampled both inert and biogenic substrates in situ, as well as artificial substrates. Biogenic substrates sampled included the surface of the red alga Prionitis sternbergii (n = 2), the anemone Anthopleura elegantissima (n = 2), and gill tissue (n = 3) and siphon tissue (n = 1) of the California mussel Mytilus californianus. These biogenic hosts were chosen because they are persistent members of the community and have relatively long-lived and sessile tissues that might provide a predictable substrate. In addition to the biogenic surfaces, rocks were chipped off of bench adjacent to mussel beds. Artificial substrates (glass crucible covers, 3 cm diam, www.leco.com) were attached with epoxy on rock adjacent to mussel beds (n = 4) and in tidepools (n = 5) on 10 Jun 2009 and left in situ for 2 months. We chose glass crucible covers because they provided heterogeneous surface, yet an inert substance. Immediately after a late morning collection at low tide, samples were frozen and sent to −80 °C storage at the University of Chicago prior to extraction. Environmental characteristics of the seawater, including nutrient concentrations, were recorded as part of a regular sampling program (Wootton & Pfister, 2012) and are reported in Table S1. The Upwelling Index for this latitude (48°N) was, on average, positive (www.pfeg.noaa.gov).

We tested whether the presence of mussels had entrained different microbial assemblages by sampling natural and artificial substrates in tidepools where mussels had been removed or unmanipulated since 2002 at Second Beach (Pfister, 2007), a north-facing complex of rocks 2 km east of Neah Bay, WA, USA within the Makah Tribal Reservation. For seven years prior to our microbial sample collection, mussels have been excluded from 6 tidepools that previously contained mussels by pulling them out by hand at an approximately monthly intervals during the spring and summer months. Other tidepools that naturally had mussels served as controls. The biogeochemistry of these experimental and control tidepools have been characterized multiple times, allowing us to test for mussel effects on tidepool nutrient concentrations (nitrate, nitrite, ammonium, and phosphorus), and seawater pH and oxygen and how it related to microbial community structure. These parameters were measured 9 times in each tidepool during Aug 2009 and 12 times in Jul and Aug of 2010. We also estimated ammonium remineralization and removal rates with a tracer experiment with enriched 15NH4+ in 2010 (Pather et al., 2014). A ∼2 cm2 piece of rock was chiseled from 5 pools with and 5 without mussels during morning hours on 24 Aug 2009. We also sampled from glass cover slips that had incubated for 3 months in these pools (n = 7 controls, n = 6 mussel removal pools) with a copper paint barrier to exclude molluscan grazers. To compare this benthic microbial assemblage to microbes in the water column, we collected 3 plankton samples on 25 mm GF/F filters by pumping 300 mL of seawater on an incoming tide (1100–1130 h). Due to the 0.7 um pore size, we likely excluded many free-living microbes, though high species richness still resulted (see below). All samples were frozen and sent to storage at −80 °C at the University of Chicago prior to extraction.

For DNA extraction, we used the Power Soil DNA Extraction Kit (MoBio). The rock, crucible cover substrates, and mussel shells were both swabbed with sterile cotton applicators and brushed with sterile spiral dental brushes that were placed in the extraction solution. Coverslips, filters, excised mussel gill or mussel siphon, anemones, and algae were placed into the beadbeater vial and pulverized. Thus, our sampling of biogenic substrata included the potential that microbes were part of the host tissue. The PCR amplification protocol followed Caporaso et al. (2010b) for multiplexing 16S rRNA samples. The PCR products were cleaned with MoBio™UltraClean htp PCR Clean-up kit. We amplified the V4 variable region of the 16S rRNA gene from community DNA using bar-coded primers according to Earth Microbiome Project standard protocols (www.earthmicrobiome.org).

Sequence analysis

We used QIIME (v. 1.7.0, Quantitative Insights into Microbial Ecology; www.qiime.org) to filter reads and determine OTUs as described in Caporaso et al. (2010b). Briefly, we used the open reference OTU picking workflow, where sequences were first clustered with the Greengenes (Dec 2012) reference database (McDonald et al., 2012); we then allowed OTUs that did not cluster with known taxa (at 97% identity) in the database to cluster de novo. Singleton and chloroplast-derived sequences were removed prior to downstream analyses. Representative sequences for each OTU were aligned using PyNast, with a minimum alignment overlap of 75 bp (Caporaso et al., 2010a). Alignments were used to build a phylogenetic tree (FastTree; Price, Dehal & Arkin, 2009). We computed alpha diversity metrics among substrates using the alpha_diversity.py script in QIIME (chao1, phylogenetic diversity and equitability), using the same sequence depth for all samples (50,000 sequences per sample). We used the beta_diversity_through_plots.py script to compute beta diversity distances between samples (weighted UniFrac), and to construct principal component (PCoA) plots, thus accounting for both the phylogenetic composition (Lozupone et al., 2011) and the relative abundance of taxa. To test for significant sample groupings based on these distance metrics, we employed PERMANOVA and PERMDISP using the compare_categories.py script in QIIME. We tested whether the abundance of particular OTUs differed significantly among different substrates using ANOVA analyses (Bonferonni corrected) with the otu_category_significance.py script. OTU networks were constructed using the make_ otu_network.py script in QIIME. We further visualized the extent to which OTUs were shared or unique among samples using Cytoscape network layouts (www.cytoscape.org). Finally, we tested for patterns in species co-occurrence as a function of mussels with a checkerboard score (c-score) analysis using the oecosimu function (Vegan package) with the ‘quasiswap’ method (99 simulations) for null model construction (Barberán et al., 2012).

Because nitrogen metabolism in association with animals was demonstrated in these locales (Pather et al., 2014; Pfister, Altabet & Post, in press), we tested whether taxa known to be involved in nitrogen metabolic pathways were present using 3 methodologies. We first examined the taxa identified (down to the level of genus) with the Greengenes database and 16S data in each sample. From literature reports, we assigned taxa to one of 4 transformations: nitrification (either ammonia oxidation or nitrite oxidation), anammox, or nitrate reduction via DNRA or denitrification based on genera associated with each metabolism (Table 1), comparing categories in R (version 2.15, www.R-project.org).

Table 1 The genera associated with different microbial nitrogen transformations.

The genera associated with different microbial nitrogen transformations that were searched in all samples via the Greengenes database. For nitrification, any taxa associated with ammonium or nitrite oxidation are included, while nitrate reduction includes taxa for denitrification and DNRA. For anammox, we searched for taxa with “Candidatus” status.

Nitrification	Nitrate reduction	Anammox	
Alcaligenes	Azospirillum	“Candidatus”	
Nitrobacter	Campylobacter		
Nitrococcus	Desulforvibrio		
Nitrolancetus	Nitratifractor		
Nitrosococcus	Nitratiruptor		
Nitrosolobos	Paracoccus		
Nitrosomonas	Rhodobacter		
Nitrosopumilis	Sulfospirillum		
Nitrosovibrio	Wolinella		
Nitrospina	Vibrio a		
Nitrospira			
Nitrotoga			
Paracoccus			
Notes.

a Due to the abundance and functional diversity of the Vibrio genus and the possibility that many Vibrio species are not involved in nitrate reduction, we did not include Vibrio OTUs in Tables 2 and 3.

Our second analysis of potential nitrogen cycling used PICRUSt to predict the percent of sequences associated with nitrogen metabolism in our 16s data (Langille et al., 2013). Briefly, all OTUs not assigned to the Greengenes database were removed from the OTU table (closed reference), abundance was normalized by 16s rRNA read number, and PICRUSt metagenome predictions were calculated (Greengenes May 2013 release). By inferring the gene families present in our 16s data, we compared whether the different living and inert substrates on Tatoosh Island differed in the amount of OTUs associated with nitrogen metabolism or whether inert substrates in tidepools with or without mussels differed in the abundance of taxa related to nitrogen metabolism.

The discovery of nitrogen metabolizing taxa with 16S rRNA data require that those taxa are described and identical or analogous sequences are available. In contrast, shotgun metagenomics directly identifies the sequence associated with metabolic function. Two of our samples of M. californianus shells were also shotgun pyrosequenced (Pfister, Meyer & Antonopoulos, 2010), which allowed us to compare nitrogen metabolisms detected with shotgun metagenomics with 16S OTU reads and PICRUSt predictions.. We sequenced the 16S rRNA V4 amplicons using the Illumina platform described above from DNA archived from a previous extraction from Tatoosh Island, where microbial community biofilms associated with mussel shell surfaces were extracted and metagenomes sequenced using 454 GS-flx pyrosequencing (Pfister, Meyer & Antonopoulos, 2010). Briefly, 6 mussels were collected on 10 April 2008 from 6 tidepools and 6 additional mussels approximately 5 m apart on an adjacent exposed rocky bench. All shells were immediately cleaned of all soft tissue. Mean shell length was 4.47 cm and 4.42 cm for tidepool and bench mussels, respectively. We thus compared the 16S rRNA V4 amplicon Illumina sequences to the shotgun metagenomic pyrosequencing to determine overlaps in key taxa. The two metagenomes from Pfister, Meyer & Antonopoulos (2010) were reanalyzed using MG-RAST for all taxonomic matches and using the SEED Subsystems database for nitrogen metabolism with maximum e-value of e < 10−5 (Meyer et al., 2008 http://metagenomics.anl.gov/).

Results

Some distinction exists among microbial assemblages associated with different substrates at Tatoosh Island

Between 54,490 and 250,432 sequences per sample were generated for 26 samples from a range of materials including inert surfaces (rock and glass crucible lids) as well as mussel shells and tissues, algal fronds, sea anemones, and the filtered plankton. All samples were rarified to 50,000 sequences per sample. OTU richness (total diversity) estimates were greatest for inert substrates and the water column, while the lowest richness was associated with mussel gill and siphon tissue (ANOVA, F7,16 = 4.968, p = 0.004). Species richness was highly correlated with other metrics of diversity including chao1 and phylogenetic diversity (r = 0.98 to 0.99, p < 0.001), as was equitability (or evenness, r = 0.780, p < 0.001). Alphaproteobacteria, including Rhodobacteraceae and Hyphomonadaceae, dominated the algal Prionitis tissue and the inert substrates, while Gammaproteobacteria, especially Vibrionaceae, dominated mussel gill and siphon tissue (Fig. 1). The communities associated with mussel shells, anemones, and filtered plankton samples were similarly dominated by Gammaproteobacteria. Mussel shells and anemones similarly had many Vibrionaceae OTUs, with shells also harboring Moritellaceae. OTUs in the Psychromonadaceae were prominent in the plankton. Given our use of a 0.7 µM filter, the OTU richness may be underestimated if the smallest bacteria were not retained.

Figure 1 The proportional representation of OTUs and the mean observed OTU richness among substrates sampled.

The proportional representation of OTUs among the major microbial groups (colored bars), with the overall mean observed OTU richness (+SE) among all substrate types at (A) Tatoosh Island, and (B) in tidepools where natural rock substrate and coverslips were sampled in the context of an experimental removal of mussels. The substrates in (A) showed significant differences in observed richness (ANOVA, F7,16 = 4.968, p = 0.004) with rocks (n = 3), crucible lids (n = 9) and filtered plankton (n = 3) showing the greatest richness while the lowest observed richness was associated with mussel gill (n = 3) and siphon (n = 1) tissue. OTU richness of mussel shell (n = 2), anemone (n = 2), and red algae (n = 2) was intermediate to the others. In (B) Tidepools with mussels removed had greater OTU richness than those with mussels (Two-Way ANOVA, F1,18 = 12.759, p = 0.002) while rock had over twice the OTU richness of coverslips (F1,18 = 140.59, p < 0.001); there was no interaction between substrate and mussel presence.

In addition to differences in alpha diversity, the microbial community composition and structure on different biotic and inert substrates showed differences in beta diversity. First, the same substrates clustered in a PCoA analysis based on weighted UniFrac distances (Fig. 2), e.g., rock substrates clustered with the glass crucibles, while mussel gill and siphon tissue clustered together. The filtered plankton samples were highly similar to each other, while the anemone and Prionitis tissues suggest greater differences among individual hosts. Substrate differences were significant with a permuted ANOVA (F5,18 = 6.570, p < 0.001) when we grouped substrates into 6 categories (anemone, red alga, plankton, mussel shell, mussel internal tissue, and inert substrates). Analysis of the differences among those 6 categories showed that each differed significantly from one or several others, except for filtered POM, which did not statistically differ from any other group.

Figure 2 A PCoA of the OTU beta diversity of substrates on Tatoosh Island.

A PCoA of the OTU beta diversity of substrates on Tatoosh Island, demonstrating the clustering among the different microbial assemblages associated with each substrate. The weighted UniFrac metric was used to incorporate relative abundance; the first axis explained 40.2% of the variance, while the second explained 14.8%. Differences among substrates were significant (PERMANOVA, F5,18 = 6.570, p < 0.001), and groupings that included anemone, Prionitis, mussel shell, mussel tissue, and inert substrates were differentiated while plankton were indistinguishable from all.

A second test indicating beta diversity differences among substrates was revealed in an ANOVA on OTU abundance. There were 10 OTUs that differed significantly in abundance (Bonferroni corrected ANOVA, p < 0.05, Fig. 3) and these distinctions came primarily from their abundance in association with macrobiota. For example, Moritella and Aliivibrio were found on mussel shell and gill tissue, while Cyanobacteria were primarily associated with algal fronds.

Figure 3 The relative abundance of the 10 OTUs that differed among the Tatoosh Island substrates.

(Boniferroni-corrected ANOVA, p < 0.05).

Further, the beta diversity differences we detected among substrates were a function of the abundance of OTUs among samples. By comparing OTUs shared between substrates in a network plot, it was evident that the abundant OTUs that were detected at least 5,000 times were generally common to all substrates (Fig. 4A). In contrast, when we examined shared diversity among rare OTUs (those detected only 5 to 10 times across the dataset, Fig. 4B), there was strong differentiation among substrates, indicating that the rare OTUs were responsible for the majority of the compositional differences between the microbial communities associated with different substrates.

Figure 4 Shared OTU diversity among microbes sampled from the substrate groupings at Tatoosh Island and portrayed as a spring-embedded layout.

Shared OTU diversity among microbes sampled from the substrate groupings as in Figs. 1–3 at Tatoosh Island and portrayed as a spring-embedded layout, where OTUs that are in common bring nodes or samples together and OTUs that are distinct repel nodes. In (A) only common OTUs detected more than 5,000 times are included, while (B) shows only rare OTUs that were present 5–10 times across the entire dataset.

The presence of mussels has little impact on microbial assemblages in tide pools at Second Beach

In experiments performed at Second Beach, the biogeochemical parameters of tidepools were affected by the presence or absence of mussels (Table S1). A principal components analysis that included the ammonium regeneration and removal rates (Pather et al., 2014), the maximum seawater pH and dissolved oxygen, ammonium, nitrate, nitrite and phosphorus measured in the tidepools over both daytime and nighttime low tides indicated that the first principal component explained 81.7% of the variance and differed among mussel versus no mussel tidepools (p = 0.049, Fig. S1). Rock had more than twice the microbial diversity of coverslips (Rock = 3,727 OTUs, Coverslips = 1,750 OTUs; F1,18 = 140.59, p < 0.001), perhaps reflecting greater time in the environment. Both types of substrata maintained more diverse and equitable community profiles in tidepools where mussels were removed, than in those with mussels present (Rock = 3,282 OTUs; coverslips = 1,343 OTUs; Fig. 1B, Two-way ANOVA, F1,18 = 12.759, p = 0.002). However, there was no interaction between substrate and mussel presence (F1,18 = 0.013, p = 0.909), indicating that the distinction between microbial communities associated with natural rock and glass coverslip communities did not depend upon the presence of mussels. The equitability with which diversity was distributed was strongly correlated with total diversity (r = 0.920, p < 0.001), indicating that when mussels are removed, the equitability of taxa also increases. Further, the microbial communities associated with the rock substrate in tidepools at Second Beach were similar in OTU composition with rock substrate at Tatoosh Island (Fig. 1A vs. 1B).

A high degree of OTU sharing and community structure similarity were observed between microbial communities associated with rock surface or coverslip samples regardless of whether mussels were present (Figs. 5 and 6); this was supported by the absence of OTUs with significantly different relative abundances between tidepools with or without mussels based on Bonferroni-corrected ANOVAs. However, there were fewer shared OTUs among rock samples when OTUs that were rare were considered against OTUs that were common (26.7% versus 96.9% shared). There was no relationship in the degree of OTU sharing with mussel presence or absence. Indeed, in contrast to our comparison of Tatoosh Island substrates, both rock and coverslip samples showed no differentiation between common or rare OTUs as a function of mussels, and the clustering of nodes was highly similar. Further, the presence of mussels was not associated with any changes in the relative weight of deterministic and stochastic forces governing community assembly. OTU co-occurrence was significantly non-random on rock surfaces (c-score analysis; p < 0.01), and this pattern did not change with mussels.

Figure 5 A PCoA of the OTU diversity of tidepool rock versus coverslip substrates at Second Beach, WA.

A PCoA of the OTU diversity of tidepool rock (n = 10) versus coverslip (n = 12) substrates at Second Beach, demonstrating strong clustering among the microbial assemblages from the two substrates, while the presence of mussels (filled symbols) versus removal of mussels (open symbols) were not a factor for explaining beta diversity. Using weighted UniFrac, the first axis explained 46.5% of the variance, while the second explained 20.3%.

Figure 6 Shared OTU diversity among microbes sampled from tidepool rock versus coverslip substrates in tidepools at Second Beach, WA.

Shared OTU diversity among (A) microbes sampled from tidepool rock versus coverslip (lighter green) substrates and (B) samples distinguished by whether mussels were present (blue) or absent (red) from tidepools at Second Beach. The spring-embedded layout shows OTUs that are in common bring nodes or samples together and OTUs that are distinct repel nodes. Only common OTUs greater than >5,000 are included. Analyses of relatively rare OTUs did not change the network pattern.

Patterns in the distribution of a subset of taxa involved in nitrogen metabolism do show responses to the presence of mussels in tide pools on Second Beach

Nitrifying taxa were at low incidence throughout our samples, while taxa related to nitrate reduction were found in almost every sample (Table 2). Filtered plankton had the highest incidence of nitrifying OTUs as a result of the genus Paracoccus. The relatively high incidence of denitrifying taxa in the anemone Anthopleura was primarily driven by matches in the Camplyobacteraceae, the group including Campylobacter, a taxon that harbors nrf genes (Pittman & Kelly, 2005). Although most samples also had genera in the Planctomycetes, only some are known to perform anammox (Fuerst & Sagulenko, 2011), and these genera were either absent from our samples or characterized as “other Planctomycetes”. We tallied these OTUs for Table 2, but did not perform statistical analyses and interpret these with caution. OTUs associated with nitrogen metabolism, as inferred from PICRUSt, were found across all Tatoosh substrates (Table 2).

Table 2 The OTUs discovered in each substrate type at Tatoosh Island associated with 3 broad nitrogen transformations.

The mean percent of OTUs discovered in each substrate type that is associated with each of the 3 broad nitrogen transformations (taxa listed in Table 1) or overall nitrogen metabolism (PICRUSt) at Tatoosh Island. No “Candidatus” were found in the 16S; the anammox category contains Planctomycetes as an estimate of anammox potential only. Mussel shell samples were analyzed with both the V4 region of the 16S rRNA as well as through shotgun metagenomics.

		Nitrification	Nitrate reduction	Anammox	N metabolism	
Substrate	Substrate type	16s	Metagenome	16s	Metagenome	16s	Metagenome	PICRUSt	
Mussel siphon	biogenic	0.0000		0.0326		0.0008		0.8811	
Mussel gill	biogenic	0.0000		0.4159		0.0009		0.9295	
Mussel shell	biogenic	0.0000	0.4604	0.0049	0.9882	0.0043	0.1711	0.8344	
filtered plankton	biogenic	0.0102		0.1851		0.0391		0.7636	
A. elegantissima (anemone)	biogenic	0.0000		1.8961		0.0804		0.8658	
Prionitis (red alga)	biogenic	0.0000		0.4278		0.0399		0.6638	
rock	inert	0.0048		0.0551		0.0390		0.7119	
crucible lid	artificial	0.0094		0.2120		0.0218		0.6325	

The percentage of taxa associated with nitrogen transformations and residing on mussel shells was compared between the 16s rRNA amplicon data and existing shotgun metagenomic data (Pfister, Meyer & Antonopoulos, 2010). The metagenomic data revealed a greater proportion of nitrogen metabolizing taxa, including taxa that were not observed in the amplicon data (Table 2). An analysis of SEED Subsystem functions for the two mussel shell metagenomes yielded estimates of 1.4% and 1.3% of the 68,676 and 63,950 proteins with functions known to be related to nitrogen metabolism that were discovered in each sample. The PICRUSt analysis, which inferred functional gene presence, indicated that 0.83% of the OTUs discovered were related to nitrogen function (Table 2), a value closer to the metagenome discovery rate than our discovery analyzing only the taxa in Table 1 with 16S rRNA data, and likely larger due to the inclusion of many nitrogen-metabolizing taxa in addition to those in Table 1.

When we compared the presence of OTUs with taxa associated with certain nitrogen metabolisms (e.g., Table 1) in our experimental tidepools, we found that the presence of mussels increased the incidence of putative nitrate reducing taxa on rock substrate, but mussels had no effect on putative nitrifiers (Table 3). The nitrogen metabolisms on inert substrates that were inferred through PICRUSt also did not differ between tidepools with or without mussels (Table 3). The maximum dissolved inorganic nitrogen in each tidepool did not correlate with the observed diversity (r = −0.318, p = 0.371). Over all 46 samples from Tatoosh and Second Beach that we analyzed, the discovery rate of OTUs with known nitrogen transformations (Table 1) was unrelated to the observed diversity (r = −0.127, p = 0.399, n = 46).

Table 3 OTUs discovered on inert substrates in experimental Second Beach tidepools associated with 3 broad nitrogen transformations.

The mean percent of OTUs discovered on inert substrates in experimental Second Beach tidepools that were associated with each of the 3 broad nitrogen transformations (taxa listed in Table 1) or overall nitrogen metabolism (PICRUSt). p-values are listed for t-tests for a significant difference on each substrate as a function of mussel presence. The only significant contrast was the greater incidence of OTUs associated with nitrate reduction on natural rock substrate in tidepools with mussels.

Tidepool substrates	Nitrification	Nitrate reduction	N metabolism	
	16s	T-test	16s	T-test	PICRUSt	T-test	
Natural rock substrate with mussels (n = 5)	0.0001	p = 0.284	0.1890	p = 0.043*	0.7433	p = 0.952	
Natural rock substrate without mussels (n = 5)	0.0011	0.0151	0.7422	
Coverslip with mussels (n = 6)	0.0004	p = 0.456	0.0737	p = 0.471	0.7421	p = 0.218	
Coverslip without mussels (n = 6)	0.0019	0.1785	0.7084	
Notes.

* indicates p < 0.05.

Discussion

All sampled substrates had microbial assemblages, which showed variation in important taxonomic and functional properties. Differences in diversity and OTU composition may reflect differences in colonization preference, temporal dynamics, the duration in the environment to accrue microbes, and host-specific interactions on the longer-lived macrobiota. The red alga, the anemone and the mussels are all long-lived and could have been present for years prior to our sampling and had ample time to accumulate a microbial assemblage. The clonal nature of the anemone Anthopleura elegantissima might even provide nearly immortal tissue for microbial proliferation. Relatively high diversity on rocks and crucible lids may indicate a number of micro-niches, perhaps as a result of surface irregularities on these substrates. The biotic substrates we sampled are especially likely to have a number of microbial niches, including microtopography and strong oxygen gradients (Heisterkamp et al., 2013).

Similarities between the filtered plankton and the animal tissue may have resulted from mussel and anemone tissue harboring planktonic microbiota, due to their feeding activities. Although increased sample size will be needed to quantify the extent of within-host heterogeneity, the animals we examined were dominated by Gammaproteobacteria, including the symbiont-bearing anemone. In contrast, coral reef invertebrates with eukaryotic photosynthetic symbionts are have been shown to be dominated by Alphaproteobacteria (e.g., Bourne et al., 2013), suggesting differential colonization drivers in different symbiont-bearing invertebrates. Nevertheless, many specialized symbionts of marine bivalves fall within the Gammaproteobacteria (Stewart & Cavanaugh, 2006; Newton, Girguis & Cavanaugh, 2008).

The great similarity in microbial community structure on inert substrates in tidepools regardless of the presence or absence of mussels (Figs. 1B, 3, 5 and 6), suggests that the enhanced nitrogen regeneration and uptake that has been demonstrated with mussels (Pather et al., 2014) could be due to the microbes on the mussels themselves. The only important functional difference that we found was a greater incidence of nitrate reducing taxa on rocks in the presence of mussels (Table 3), a result likely explained by the ability of mussels to temporarily reduce oxygen levels to the point where reducing processes are favored. OTUs associated with nitrification did not differ. Although mussel presence was associated with a decreased alpha diversity on inert substrates (Fig. 1), this did not lead to significant differences in the network of OTUs that were shared across tidepools with and without mussels and thus did not affect beta diversity (Fig. 6). Hence, the macrobiota did not drive any shifts in community structure on nearby inert substrates, even though tissue-associated microbial communities were significantly differentiated. Both 16S rRNA and shotgun metagenomic analyses (Table 2) suggest that macrobiota host OTUs that are important for nitrogen cycling. This corroborates work by Welsh & Castaldelli (2004) that showed nitrogen metabolism was hosted within a related mussel species. Similarly, deep-sea mussels are known to host nitrogen-utilizing symbionts (Lee & Childress, 1995). In contrast with the shell surface microbes, those with mussel gill and siphon were less diverse (Fig. 1). Thus, the elevated nitrite concentrations in tidepools with mussels, suggesting increased nitrification (Pfister, 2007), may be the result of a direct effect of habitat provisioning for microbes by the macrobiota, rather than simply a microbial community shift on other substrates due to nutrient provisioning. Alternatively, the nitrogen increase due to mussels may not be enough to drive microbial community differences. Indeed, neither microbial community composition nor the expression of several functional genes in coastal sediments showed major changes in response to nutrient perturbations (Bowen et al., 2011). Thus, further exploration of macrobiota as the repository of microbial function, not just as providers of nutrient resources, is warranted.

We note that nitrogen-transforming taxa were found with a higher incidence in the metagenomic data than the 16S rRNA V4 amplicon sequencing of mussel shells. Although there are technical differences between the two sequencing methodologies that could lead to detection differences, such as GC bias (Ross et al., 2013), metagenomic data directly identify genes for nitrogen transformations that are relatively conserved and identifiable (independent of host phylogeny), even if the taxa hosting these genes are uncharacterized. Many of the taxa identified in the amplicon survey are not closely related to cultured isolates, and their N-cycling status is unknown. Although this limits our ability to infer function from phylogeny, it nevertheless is an analysis that may become increasingly insightful as our knowledge of sequence-based diversity increases. While we recognize that OTU analysis using 16S rRNA data predictions may not yet be the strongest lens to detect function, the increased characterization of taxa involved in nitrogen metabolisms (Ward, Arp & Klotz, 2011; Munn, 2011), and the analyses of 16S data with PICRUSt (Table 2) suggests that functional inference is possible.

Marine microbial community structure has been shown to be composed of a multitude of rare taxa that have deep phylogenetic differences; the extent to which this ‘rare biosphere’ (Sogin et al., 2006) drives community function is unknown, though deep sequencing efforts have revealed that there could be the equivalent of a ‘seed bank’ of rare taxa that are persistent with only relative abundance changing through time and space (Lennon & Jones, 2011; Caporaso et al., 2012; Gibbons et al., 2013). In some ecosystems, rare taxa have also been shown to be as metabolically active as common species (Hamasaki et al., 2007), suggesting that rarity does not preclude functional importance. Although seawater and rock had the highest OTU diversity, the differences among filtered plankton and inert substrates versus biogenic substrates (mussels, seaweed, anemone) sampled in situ demonstrated that macrobiota enhance beta diversity by hosting unique OTUs (Figs. 3 and 4B), presumably via the provisioning of unique habitats or resources. It is possible that particular microbial taxa end up in association only with intertidal macrobiota, though the selectivity of these associations requires further temporal and spatial sampling. Our results with these benthic macrobiota are, however, in direct contrast to analyses of seawater where the patterns of beta diversity did not differ among abundant and rare taxa (Amaral-Zettler et al., 2010).

The demonstration that macrobiota host a unique microbial community compared to the water around them is supported by this study and other recent work with tadpoles (McKenzie et al., 2012) and marine algae (Michelou et al., 2013). Although it has been recognized for several decades that benthic invertebrates in deep sea environments host unique taxa e.g., DeChaine & Cavanaugh (2006), it may be that benthic macrobiota common to large parts of the ocean are also repositories for unique microorganisms (Grossart et al., 2005; Lee et al., 2011; Bengtsson et al., 2012; Jackson et al., 2012) and loci for important biogeochemistry (Martínez-García et al., 2008; Heisterkamp et al., 2013). The rocky intertidal flora and fauna, though relatively well-understood in terms of the interactions among macrobiota, likely also interact and mediate productivity via a rich microbial community that we are just beginning to describe. These intertidal macrobiota may also harbor a unique set of taxa adapted to host-associated niches, thereby promoting microbial community diversity in the coastal ocean.

Supplemental Information

Figure S1 A principal component analysis of the environmental parameters measured in the experimental tidepools at Second Beach, WA

A principal component analysis of the environmental parameters (seawater pH, oxygen, and nutrient concentrations), as well as the ammonium regeneration and removal rates (from Pather et al., 2014) measured in the experimental tidepools at Second Beach (Table S1). The first principal component explained 81.7% of the variance and differed among mussel controls (filled symbols) versus removals (open symbols, p = 0.049).

Click here for additional data file.

Table S1 The environmental parameters of experimental tidepools at Second Beach

The environmental parameters of tidepools at Second Beach where mussels were either removed or remained shown as means (se). All values were the maximum measured in each tidepool (see Pather et al., 2014). The Principal Components Analysis based on these environmental parameters is shown in Figure S1. The last column represents environmental parameters for seawater surrounding Tatoosh Island (see Wootton & Pfister, 2012) encompassing the period that artificial substrates were deployed and rock and biogenic substrates collected (10 Jun to 6 Aug 2009).

Click here for additional data file.

S Owens provided expertise in the lab, and G Caporaso and J Stombaugh provided critical code. We thank A Olson for help in the field, O Moulton for lab assistance, and M Coleman for comments on the ms. We are grateful to the Makah Tribal Council for access to their lands.

Additional Information and Declarations

Competing Interests

Author Contributions

Field Study Permissions

DNA Deposition

Data Deposition

The authors declare there are no competing interests, financial or otherwise, for publishing this paper.

Catherine A. Pfister conceived and designed the experiments, performed the experiments, analyzed the data, contributed reagents/materials/analysis tools, wrote the paper, prepared figures and/or tables, reviewed drafts of the paper.

Jack A. Gilbert conceived and designed the experiments, analyzed the data, contributed reagents/materials/analysis tools, wrote the paper, reviewed drafts of the paper.

Sean M. Gibbons analyzed the data, prepared figures and/or tables, reviewed drafts of the paper.

The following information was supplied relating to field study approvals (i.e., approving body and any reference numbers):

The Makah Tribal Nation provided written permission through the Makah Tribal Council in Neah Bay, WA.

The following information was supplied regarding the deposition of DNA sequences:

All amplicon and metadata has been made public through the Environmental Microbiome Project data portal (www.microbio.me/emp).

The following information was supplied regarding the deposition of related data:

Data on the tidepool biogeochemistry are at bco-dmo, http://hdl.handle.net/1912/6420.

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
