# Peer review of "The role of macrobiota in structuring microbial communities along rocky shores"

_PeerJ, doi:10.7717/peerj.631_

## Round 0.1 · original submission · Minor Revisions

· Academic Editor

Minor Revisions

I agree with both referees who find that the paper is novel, interesting and worthy of publication. However both have a number of suggestions for clarification and some additions prior to final acceptance of the paper. Both referees are microbial ecologists, but the first still suggests a revision of the introduction to make the paper more easily to non-specialists. The second referee notes the complexity of the design and appreciates the clarity of the writing. However, they also have some points of disagreement with the interpretation or presentation of the results that are important and should be addressed in the revision. In particular, they question the degree of replication and a number of conclusions they feel are unsupported (in particular community distinctiveness and OTU vs species), which I agree should be dealt with appropriately during revision. Both referees are nonetheless supportive of the manuscript and I find myself in agreement with them that a minor revision that includes appropriate caveats about the issues of replication, sequencing depth and community distinctiveness should be acceptable for publication.

Reviewer 1 ·

Basic reporting

I thank the authors for the opportunity to review this important body of research.

According to PeerJ’s Basic Reporting requirements, “The article should include sufficient introduction and background to demonstrate how the work fits into the broader field of knowledge.” I think the Introduction section needs to be reorganized and additional information added for this manuscript to fulfill those requirements. In general, the Introduction does not flow readily, making it difficult for the reader to understand the logic behind the study, and is not strong in setting the stage for the hypotheses addressed in this study. I suggest the following in order to remedy these issues:

The first and second paragraphs of the Introduction are somewhat redundant and contain repetitive information. They should be integrated, condensed, and reorganized to improve the flow of the text. For example, Lines 33-35 and Lines 43-44 should/could be adjacent to one another - perhaps moved the second paragraph where authors can focus on distinguishing potential differences in microniches between upwelled seawater and macrobiota surfaces. Most of the information is there; it just needs needs to be restructured.

Lines 29-33: Delete this sentence. It disrupts the logical flow of the paragraph and does not appear to be critical background information for the introduction section.

Lines 37 and 41: The word “nascent” used close together here. May want to choose something else.

Perhaps, it would be beneficial to elaborate (with references) on the significance of how microbial communities on living vs. non-living substrates may be contributing to biogeochemical cycles in coastal ecosystems. Frame your hypothesis for the reader. The authors cite the "what" but not the "why?"

Experimental design

The large pore size of filters used to sample bacterioplankton is of concern (0.2 micron is the standard in microbial ecology). It is mentioned in the methods, but not addressed in the discussion. As the authors admit, they “likely excluded many free-living microbes.” This issue needs to be very clearly stated and highlighted throughout the manuscript, or the seawater data should be removed from the comparative analyses altogether.

Methods should be clear and reproducible. Please address the following:

Lines 68-72: Authors list that microbial biofilms were collected from biogenic substrates, but Lines 107-109: state that pieces of gills, siphons, anemones, and algae were placed into tubes and pulverized. Here, you would be sampling more than the externally associated biofilm since you would be including any microbes potentially located inside host tissues. Please clarify this for the reader.

Lines 107-107: Swabbing Methods are unclear. Which biofilms were swabbed with cotton applicators? Which were brushed with sterile spiral dental brushes? Was each sample type swabbed/brushed with both? Why were two sampling treatments chosen? Sample treatment should be standardized amongst samples, and at the very least, it should be made clear as to which sample type was subjected to a particular sampling treatment.

Please clarify and distinctly explain in the methods section the different analyses (16S vs. metagenomic vs. PICRUSt) used to identify nitrogen metabolic pathways. The presence of three total analyses was not evident until the results section.

Validity of the findings

Overall the authors did a nice job of discussing their findings and summarizing the study.

Suggestions:

Lines 270-272: A few citations are needed here.

Lines 273-275: As opposed to the longevity (long existence) of rocks? The logic of this statement is not clear.

276-278: Here I would argue that living tissue (i.e. anemones) would also provide a large diversity of microniches for microbes to proliferate. Perhaps the authors might focus on highlighting differences in the types instead of number of microniches for living vs. nonliving substrates leading to differences in microbial composition.

Comments for the author

Line 77: Extra space after period here?

Fig 4. And Fig. 6 need to be labeled – including samples and graph headings (e.g. common vs. rare)

Is the supplementary info attached?

Tables should be formatted consistently

Line 283: Is the comma supposed to be a period?

Line 304: Microbes “in” or “on” (or both) mussel gill and siphon? Please clarify and keep consistent throughout the manuscript.

·

Basic reporting

This manuscript deals with the microbial assemblages on rocky shores, and how they differ between some of the major microbial habitats in this environment, including living surfaces, the water column and inert surfaces. The study design focuses on a field site at Tatoosh Island, and includes both natural and experimentally manipulated samples. Experimental removal of mussels from tidepools serves to shed light on nitrogen metabolism of the microbial communities in this environment.
I like the holistic approach to the rocky shore environment, rather than studying individual host/substrate-microbial systems, which has been more common until now. The manuscript also has a good balance between discussion of structural (community composition) and functional (nitrogen metabolism) aspects. I think this is an important contribution. The manuscript is generally well written and quite concise considering the complexity of the experimental samples.

Experimental design

The experimental design suffers from some shortcomings relating to replication (see below), which would not be fatal as long as the conclusions are drawn with care. It is important that some unsupported statements (see comments below) are removed before publication, or that the statements are backed up with statistical analyses.

Validity of the findings

Abstract, line 4: “ ... we show that microbial composition was significantly different between inert surfaces, the biogenic surfaces ... , and the water column plankton”. There is no mention of a test in the results section that actually shows this. Figure 2 suggests that this is the case, but it must be supported with some kind of test to call it “significantly different”. Actually, in the results and discussion sections, there are statements about the distinctness of all substrates, which is also not supported (see below).
Page 7, line 140: I do not quite understand how the taxa were assigned to nitrogen metabolism function. Did you manually assign genera to function based on information in the literature, or does greengenes provide some kind of functional information? Please specify this.
Page 7, line 145: By “non-Greengenes OTUs”, do you mean OTUs that were not assigned to a genus in Greengenes? Please explain more clearly.
Page 7, line 146: “16s rRNA copy number”. Don’t you mean “16s rRNA read number”? Copy number sounds like you have used qPCR to quantify gene copies. Please consider rephrasing to avoid confusion.
Page 7, line 148: “Tatoosh substrates” sounds a bit strange. Consider rephrasing to “The different living and inert habitats sampled on Tatoosh island” or similar.
Page 7, line 155: “.. methods of nitrogen discovery” is not a very precise formulation. Rephrase to “ .. methods of detecting nitrogen metabolism functions” or similar.
Page 7, line 156: “.. a previous extraction of Tatoosh Island” sounds like the whole island was extracted. Replace “of” with “from”.

Page 8, line 169: “ The microbial assemblages associated with different substrates are distinct”. I do not agree that this statement is justified by your data as it is presented. “Distinct” implies that each substrate/habitat has its own unique assemblage. Due to the lack of appropriate statistical replication (n≥3) for the alga, the anemone, mussel shell and mussel siphon, you are not able to show that these substrates have distinct microbial assemblages. You could perhaps show that the mussel substrates (in total n=6) are distinct from the plankton (n=3) and the inert substrates (n=11). Alternatively, all living substrates (total n=10) may be distinct from the plankton and inert substrates. This is indeed suggested in the abstract. Although it looks like one or both of these hypotheses might hold true if you look at figure 2, no statistical test to support this is presented. I suggest that you perform an appropriate statistical test (for example, see anova.cca help file in R:vegan) to show which of the (replicated) groups are distinct. The quoted statement should be removed or rephrased to more precisely describe which groups are separable in this study.
Page 8, line 173, 175: OTU richness, not “species” richness is the correct term.
Page 12, line 269: Again, the experimental setup does not allow to conclude that “All sampled substrates had distinct microbial assemblages” (see previous comment). Be honest about the limitations of the experimental setup, it is ok to speculate that the alga and the anemone probably also have distinct microbiomes based on existing literature etc. However, the experiment can’t show this, which is not a big deal as long as no false claims are made.
Page 17, References: The references contain some typos and errors, please go through them carefully. One example: Bengtsson et al. is from 2012, not 2013. The kelp is called “Laminaria hyperborea” not “Laminaria hyperborean”. ;)
Page 25, Figure 1 figure legend: “OTU diversity” is supposed to read “OTU richness” I believe? Or “rarefied OTU richness”?
Page 33, Table 1: Under the anammox column, “Candidatus” is not a genus. To my knowledge, all anammox performing planctomycete genera have candidatus status, and Scalindua is one of them. Isosphaera is not an anammox genus as far as I know. Please critically review the selection of all the genera in this table, as the selection process appears error-prone.

Comments for the author

see above

---

## Round 0.2 · accepted · Accept

· Academic Editor

Accept

I feel that you have addressed the comments of the referees and the revisions requested by each were relatively minor to start with. Given the positive reviews and the reasonable revisions to address each, I do not see any reason to delay the paper by sending it back out for additional review. I am happy to accept your manuscript, and I look forward to seeing the paper come out in the journal.